# Data-Driven Uncertainty-Aware Forecasting of Sea Ice Conditions in the Gulf of Ob Based on Satellite Radar Imagery

## Abstract

The increase in Arctic marine activity due to rapid warming and significant sea ice loss necessitates highly reliable, short-term sea ice forecasts to ensure maritime safety and operational efficiency. In this work, we present a novel data-driven approach for sea ice condition forecasting in the Gulf of Ob, leveraging sequences of radar images from Sentinel-1, weather observations, and GLORYS forecasts. Our approach integrates advanced video prediction models, originally developed for vision tasks, with domain-specific data preprocessing and augmentation techniques tailored to the unique challenges of Arctic sea ice dynamics. Central to our methodology is the use of uncertainty quantification to assess the reliability of predictions, ensuring robust decision-making in safety-critical applications. Furthermore, we propose a uncertainty-aware model switching mechanism that enhances forecast accuracy and model robustness, crucial for safe operations in volatile Arctic environments. Our results demonstrate substantial improvements over baseline approaches, underscoring the importance of uncertainty quantification and specialized data handling for effective and reliable sea ice forecasting.

# INTRODUCTION

The Arctic region is experiencing an unprecedented rate of warming, leading to a significant reduction in sea ice area by more than 30% over the last four decades, and a simultaneous decrease in sea ice thickness (Kwok, 2018). Alongside this, the last century has seen active development of ice-breaker construction technologies, including nuclear-powered ones. These changes have opened up new sea routes, such as the Northern Sea Route, which provide faster and more economical transport. However, increased navigation is accompanied by increased risks due to ice jams, posing a serious threat to maritime safety.

Traditional sea ice models, based on elastic-visco-plastic rheological properties, often fail to accurately reflect all the nuances of ice deformation, rendering them unreliable for forecasting in some cases (Nummelin et al., 2016; Eastwood et al., 2020; Li et al., 2021; Overland & Pease, 1988). Additionally, these models require significant computational resources to adequately simulate the interactions between the ocean and ice. Consequently, there is a need to explore alternative methodologies that leverage statistical methods such as machine learning techniques, known for their flexibility and lower computational demands.

Our research is aimed at improving the forecasting of ice conditions in the Gulf of Ob, a region significantly influenced by the interaction of the saline waters of the Kara Sea and the fresh water of the big northern rivers, leading to complex ice formation dynamics (Weatherly & Walsh, 1996; Osadchiev et al., 2021).

We utilize radar images obtained in the Sentinel-1 mission (Sentinel-1), weather observation data (Weather & Climate), and operational forecasts and reanalysis from the GLORYS project (GLO) to predict future sea ice conditions. From a machine learning perspective, the series of satellite radar images can be treated as a continuous video sequence, therefore the problem can be formulated as a conditioned video prediction task — the widely investigated problem in common-life domain (Ming et al., 2024). Our research employs following video prediction models:

- Implicit Stacked Autoregressive Model for Video Prediction (IAM4VP) (Seo et al., 2023) uses a fully convolutional neural network with an implicit multi-input-single-output workflow, achieving state-of-the-art accuracy of weather predictions in datasets such as SEVIR;

- Dynamic Multi-Scale Voxel Flow Network (DMVFN) (Hu et al., 2023) utilizes voxel flow for video prediction, addressing efficiency and adaptability in handling diverse motion scales;

- MotionRNN (Wu et al., 2021) models spacetime-varying motions using the Motion Gating Recurrent Unit and Motion Highway mechanisms, enhancing prediction accuracy in dynamic scenarios;

- Neural Ordinary Differential Equations (Neural ODE) and Vid-ODE (Park et al., 2021) treat consecutive frames as solutions to systems of ordinary differential equations, offering control over visual attributes and smooth transitions between frames.

As the primary loss for training models and metric for evaluating their performance, we use Mean Squared Error (MSE) between predicted and target Synthetic-Aperture Radar (SAR) images. In addition to this, we utilize the Structural Similarity Index (SSIM) (Wang et al., 2004) and its extension, the Multi-Scale Structural Similarity Index (MS-SSIM) (Wang et al., 2003), to assess the perceived quality of digital images and videos. Finally, the Integrated Ice Edge Error at level $c$ (IIEE@c) (Goessling et al., 2016) is utilized to measure the similarity between forecasted and observed ice sheets. These indicators allow us to meticulously compare the accuracy of predicted ice conditions against observed data, ensuring that our models reflect not only general trends but also detailed spatial structures necessary for accurate ice mapping.

Our contributions can be summarized as follows:

- we explore the potential of modern deep learning video-prediction models in short-term regional sea ice forecasting;

- we address the problem of data irregularity and missing values within the Arctic area by exploring filtration, normalization, and augmentations techniques;

- we show the ensemble of ML models provides sufficient uncertainty estimation, and we propose novel uncertainty-aware model switching scheme that stabilizes the forecast and enhances its quality;
- finally, we assess a gap filling performance for satellite radar imagery and demonstrate the superiority of our method in comparison with a general approach for interpolating video sequences.

RELATED WORKS

Several studies have demonstrated the effectiveness of machine learning in forecasting sea ice extent and sea ice concentration. Chi and Kim (Chi & Kim, 2017) pioneered in the use of deep learning for sea ice prediction. Their model employs multilayer perceptrons (MLPs) and long- and short-term memory networks (LSTMs) to capture complex relationships in sea ice data. By training the MLP- and LSTM-based models on historical data, they identify patterns for one-month predictions, outperforming traditional statistical models. This work highlights the advantages of deep learning in sea ice forecasting.

Recent research has extended the application of UNet-based models to sea ice forecasting, highlighting their versatility beyond original medical imaging applications. Fernandez et al. (Fernández et al., 2022) investigated coastal sea elements forecasting using various UNet-based architectures, including 3DDR-UNet and its enhanced versions. Their study demonstrated the effectiveness of these models in forecasting coastal sea conditions when using satellite imagery. Grigoryev et al. (Grigoryev et al., 2022) presented a recurrent UNet with a specialized training scheme that considerably outperformed persistence and linear trend baseline forecasts of sea-ice conditions in the regions of the Barents, Labrador, and Laptev seas for lead times up to 10 days. Kvanum et al. (Kvanum et al., 2024) showed that the similar approach in the Barents sea can overcome traditional numerical models at the forecasting of sea ice concentration at one kilometer resolution and 3-day lead time. Keller et al. (Keller et al., 2023) explored various UNet-based architectures for prediction sea ice extent at kilometer resolution for lead time up to 7 days. These studies revealed the potential of machine learning methods over traditional approaches for high-resolution sea ice conditions forecasting.

Several studies showcase the prospects of uncertainty-aware data-driven sea ice forecasting. Horvath et al. (Horvath et al., 2020) suggested using Bayesian logistic regression to forecast September minimum ice cover from 1-month up to 7-month lead times. In this paper Bayesian uncertainty quantification helps to assess the reliabillity of the forecasts. Andersson et al. (Andersson et al., 2021) introduced a probabilistic deep learning sea ice forecasting system called IceNet with 6-month lead time. Their system predicts monthly averaged sea ice concentration maps at 25-km resolution, outperforming traditional models by effectively bounding the ice edge. Wu et al. (Wu et al., 2022) suggested VAE-Based Non-Autoregressive Transformer as an uncertainty-aware model for long-term sea ice concentration forecast along Northern Sea Route. Also uncertainty quantification is the key motive of the conjugate problems like sea ice data assimilation (Nazanin, 2019) or sea ice concentration retrieval (Chen et al., 2023b).

One challenge in sea ice forecasting and analysis based on satellite imagery data is the presence of noise and gaps, which can occur due to limitations of satellite trajectories, instrumental errors, data losses, and environmental factors. To address this, researchers have proposed various gap-filling methods (Desai & Ganatra, 2012). They incorporate chained data fusion, multivariate interpolation, and empirical orthogonal functions to effectively fill missing data. Weiss et al. (Weiss et al., 2014) proposed an effective approach for continental-scale gaps inpainting based on nearest neighbors method and taking in account seasonality of the data. Their approach, additionally, quantified uncertainty of the filled values. Appel (Appel, 2024) introduced a deep learning approach based on partial convolutions for filling gaps in consecutive data, highlighting the promise of deep learning for satellite imagery-related tasks. These methods enhance the quality and reliability of remote observations, having potentially a wider range of applications than just sea ice analysis.

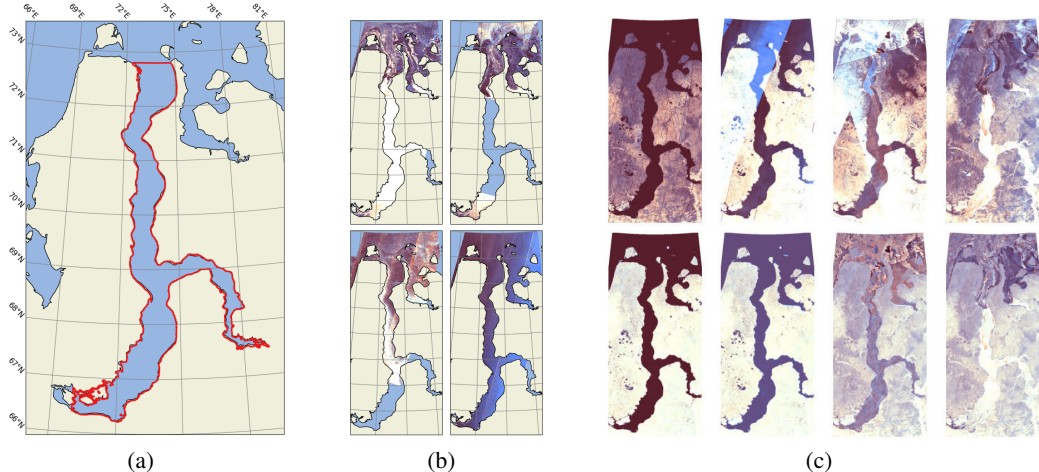

(a)             (b)             (c)

Figure 1: (a) Gulf of Ob region we are focused on. (b) Examples of colorized SAR images. (c) Examples of images before filtration (the first line) and after (the second line). Images are colorized according to sentinelhib guidelines.

## DATA

Our neural network model utilizes a number of input channels (fields) that come from three sources: Sentinel-1 (Sentinel-1) SAR imagery in HV and HH polarizations in extra-wide mode, Global Ocean Physics Reanalysis (GLORYS) (GLO), and historical data from meteostations (Weather & Climate). See detailed information in the appendix A. For the purpose of this study we set the target resolution to one kilometer, which nevertheless is sufficient enough for navigation applications (Kvanum et al., 2024; Keller et al., 2023).

The region we investigated encompasses the Gulf of Ob and the Taz Estuary in Northern Russia. The region of interest, at this one kilometer resolution, produces images with a size of $880 \times 400$ pixels. SAR images are interpolated conservatively to a covering equal-area grid in North-Polar projection (see Figure 1). GLORYS is interpolated bilinearly. Meteodata is interpolated using radial basis function interpolation. To focus on forecasting sea-ice dynamics, the land surface in target images is masked with zero values.

In this study, we selected SAR imagery as the target, due to several key advantages it offers. Firstly, Sentinel-1 allows for continuous monitoring of polar regions regardless of cloud cover or illumination. Secondly, the high spatial resolution of Sentinel-1 enables detailed analysis of sea ice, including the detection of small-scale ice features important for navigation and environmental monitoring. Thirdly, the large amount of historical data provided by Sentinel-1 is essential for training deep learning models in comparison with others sources. Finally, the almost real-time data delivery of Sentinel-1 is crucial for operational applications.

We acknowledge several disadvantages of the SAR imagery. Firstly, the revisiting period of the satellite is several days, hence many empty frames to appear when attempting to create a regular time

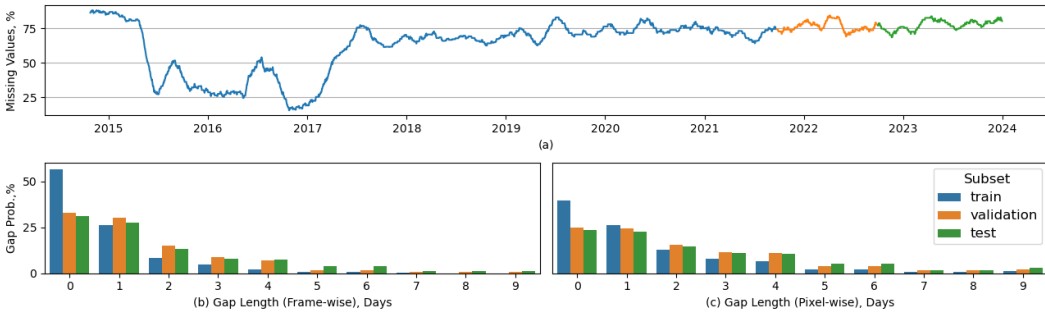

Figure 2: (a) Frequency of missing values in SAR imagery smoothed with a month-wide rolling window. (b-c) Distribution of distances between consecutive missing values across all subsets, calculated frame-wise between frames with any valid data, and pixel-wise at fixed locations.

series sequence. The distribution of missing values over subsets is depicted in Figures 2. Secondly, the entire area is not always captured in the images, resulting in some images being incomplete. Thirdly, thermal noise and imagery artifacts at HV polarization are significant, leading to varying brightness in repetitive patterns known as scalloping.

## METHODS

### DATA PREPROCESSING AND FILTRATION

The origin of the noise in SAR imagery is thermal interference within radar systems, influenced by the technology utilized for surface scanning, resulting in presence of speckles and scalloping patterns (Singh et al., 2021). Thermal artifacts have significant magnitude relative to useful information, which leads to huge biases and corrupts optimization convergence of neural networks. Therefore, we preprocess data to filter out imagery artifacts. The results of the final filtering are presented in Figure 1c.

Our custom filtering algorithm treats images as vectors from $\mathbb{R}^{H \times W}$ space with standard scalar product, where $H$ and $W$ stand for sizes of the input frames. The core assumption is the orthogonality of artifacts $A$ to the subspace of clear images $C \perp A$. Therefore, the filtering process is an orthogonal projection: $P : \mathbb{R}^{H \times W} \to C$, $P^2 = P = P^T$.

However, the construction of such operator requires the retrieval of aforementioned subspaces. The key thought is that all the ice-free frames $IF$ must have the same projection: $\exists c_0 \forall c \in IF : P(c) = c_0$. To achieve this, the frame $c_0$ with neither ice nor noise should be chosen by hand. We obtained several candidates for such a frame through visual assessment and peaked the pixel-wise minimum of all of them. Then artifact subspace $A$ is constructed from $IF$ to match orthogonality condition to $c_0$ at least, and the filtering operator $P$ is constructed after choosing a basis in the subspace containing all the artifacts $A$:

$$A = \{v - \frac{(v, c_0)}{(c_0, c_0)} c_0 | v \in IF\}, \quad P(v) = v - \sum_{i=1}^{n} \frac{(v, e_i)}{(e_i, e_i)} e_i \tag{1}$$

where $\{e_i\}_{i=1}^{n}$ is a basis in the linear span of $A$.

### VIDEO PREDICTION MODELS

To determine the relative quality of our models performance, we compare them against two baselines: persistence forecast and linear one. To obtain the parameters of the linear transformation we utilize the same techniques as for deep learning models.

IAM4VP (Seo et al., 2023) is fully convolutional neural network that leverages the trade-off between temporal-consistency of autoregressive methods and error-independence of non-autoregressive ones via implicit Multi-Input-Single-Output workflow. Like non-autoregressive methods, stacked autoregressive approach uses the same observed sequence to estimate future frames. However, the model uses its own predictions as input, similar to autoregressive methods. As the number of time steps increases, predictions are sequentially stacked in the queue. After the iterative process is finished, all generated frames are refined by the last few layers to raise the temporal correlations.

DMVFN (Hu et al., 2023) is a video prediction model leveraging voxel flow estimation to focus on movement and to handle the occlusion effect. DMVFN also incorporates a dynamic routing module that adaptively selects sub-networks based on the input frames, allowing it to handle diverse motion scales efficiently. The model's architecture includes Multi-scale Voxel Flow Blocks (MVFBs) that capture large motions and iteratively refine voxel flow estimates. DMVFN demonstrates improved efficiency and adaptability, particularly for videos with complex motion patterns and is considered a state-of-the-art deep learning solution for video prediction.

MotionRNN (Wu et al., 2021) is a model designed for video prediction, specifically addressing the challenge of predicting continuous spatio-temporal dynamics. MotionRNN is a successor of the LSTM-based architectures that also incorporates warp transformation and introduces the concept of breaking down physical motions into transient variation and motion trend. Transient variation

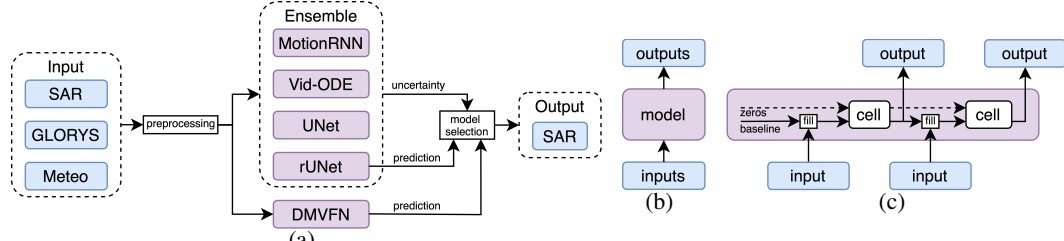

Figure 3: (a) The overall pipilene: data gathering, interpolation, normalization, filtration, neural networks evaluation, uncertainty quantification and the model selection. The final prediction is chosen from outputs of rUNet and DMVFN, based on the uncertainty, estimated by ensemble spread. (b) Non-autoregressive models treat time data as image channels and predicts output at fixed number of lead times. (c) Autoregressive models make forecast day by day, the intermediate forecast is used for missing values imputation, while the missing values at the first timestamp are imputed with persistent baseline.

represents immediate changes, while the motion trend captures the overall direction or tendency of movements over time.

Neural ODE (Chen et al., 2018) offers a powerful framework for modeling dynamic systems by means of machine learning. The core idea is to model the transformation of data through a continuous dynamic equation in a Cauchy formulation instead of discrete layers used in traditional neural networks. The forward pass is a numerical solution of a parametrized ODE. To train Neural ODEs through backpropagation the gradients can be computed either directly through the dynamic equation using automatic differentiation or more memory-efficiently through the adjoint method. The adjoint method treats gradients as solutions of a reverse-time differential equations, integrating it backwards in time.

Vid-ODE (Park et al., 2021) represents Neural ODE in a latent space on motion-vector dynamics with a warp-correction mechanism. The main idea of Vid-ODE lies in the parameterizing visual attributes such as pixels position by utilizing differential equations. The encoded input frames are assimilated into an evolving state by the GRU cell. Neural ODE models latent dynamics. The predicted state is decoded into pixels relative shifts and correction features that are used to correct warping impurities and model color and brightness variation. This iterative process ensures smooth transitions between generated frames.

UNet (Ronneberger et al., 2015), is an convolutional neural network which includes an encoder for capturing context and a decoder for precise localization for the output. The decoder path involves upsampling of the feature maps and concatenates them with the corresponding feature maps from the encoder path. Originally designed for the biomedical image segmentation, UNet has been adapted for different geophysical fields forecasts such as: coastal sea elements (Fernández et al., 2022), precipitation (Kaparakis & Mehrkanoon, 2023; Trebing et al., 2021), and sea ice concentration (Grigoryev et al., 2022; Kvanum et al., 2024). We use it within both autoregressive and non-autoregressive approaches. The former one we will mention as rUNet, where 'r' stands for recurrent.

### DATA SPLIT

The data is divided into three sets. Training set: September 1, 2015, to September 23, 2021. Validation set: September 24, 2021, to September 30, 2022; Test set: October 1, 2022, to September 30, 2023.

### IMPLEMENTATION AND TRAINING

All models are implemented in PyTorch and trained from scratch with AdamW optimizer (Paszke et al., 2019). The loss function is a combination of MSE and SSIM losses:

$$L = \text{MSE} - 0.2 \cdot \text{SSIM} \tag{2}$$

Table 1: Model configurations. Regime abbreviations are constructed as follows: SI stands for Single-Input, MI — Multi-Input, SO — Single-Output, and MO — Multi-Output, based on sequence lengths: Single-Input models acquire input iteratively, Multi-Input — at once; Single-Output models are autoregressive predictors; Multi-Output — non-autoregressive. Computational costs for each model (in GFLOPS) are provided per one input sequence. ODE-based models use adaptive solvers; the adaptive time step leads to varying GFLOPS; its standard deviation is provided.

| Model | Regime | Input size | GFLOPS | Params |
|---|---|---|---|---|
| Persistence | - | - | - | 0 |
| Linear | SISO | 7 | 33 | 1.06 K |
| DMVFN | MISO | 7 | 198 | 3.56 M |
| IAM4VP | Implicit MISO | 10 | 76 | 27.8 M |
| Neural ODE | SISO | 7 | $200 \pm 100$ | 18.53 K |
| MotionRNN | SISO | 7 | 10610 | 6.84 M |
| Vid-ODE | SISO | 7 | $480 \pm 150$ | 469 K |
| UNet | MIMO | 7 | 559 | 31.10 M |
| rUNet | SISO | 7 | 4780 | 31.04 M |

The initial learning rate is set to $10^{-3}$ and exponentially decreasing with factor $\gamma = 0.99$. The batch size is set to 32. Models are trained until either convergence of validation metrics or the overfitting begins (early-stopping). Models with the best validation score are evaluated after on the test-subset.

To mitigate bias on missed parts of the SAR input, normalization layers were removed from encoders of IAM4VP and UNet. For other models missing values are imputed with previous forecast from autoregressive prediction (see the schematic representation on Figure 3).

While training Neural ODE and Vid-ODE models, naive implementations of the adjoint method might suffer from inaccuracy in reverse-time trajectory computation, therefore in our work we have used specific implementation called MALI (Zhuang et al., 2021) that guarantees accuracy in gradient estimation.

To determine the relative quality of our models performance, we compare them against two baselines: persistence forecast and linear one. To obtain the parameters of the linear transformation we utilize the same techniques as for deep learning models. The overall models configurations are provided in Table 1.

AUGMENTATIONS

To prevent overfitting and improve generalization ability we utilize geometrical augmentations: random horizontal flips with a probability of 0.5 and uniformly sampled random rotations with angles in range $[-10°, 10°]$ with the corresponding rotation of wind and sea-currents field. To leverage the imbalance of missing values depicted at Figure 2 we utilized frameout augmentation. Up to three random frames in the input sequence are cut out until the concentration of missing values reaches the level of the test subset (70%).

UNCERTAINTY-AWARENESS

Estimating uncertainty in data-driven weather forecasting models is crucial for better model interpretation and decision-making. If the uncertainty estimation is well-calibrated, the reliable predictions are characterized by high confidence. On the other hand, low confidence means the prediction can not be trusted. In such cases one could replace it with a simple baseline or a more robust model. Following this principle, automatized pipelines of uncertainty-aware model mixture can be designed (Lakshminarayanan et al., 2017; Chen et al., 2023a; Zeng et al., 2023; Jiang et al., 2023). The mechanism is as follows: the expert model makes a prediction, its uncertainty is estimated; if the uncertainty exceeds the preset threshold, the prediction is replaced by more stable baseline. In our work the threshold is selected on the validation subset. This helps to exclude unreliable forecasts and enhances the overall performance of the forecasting system.

Traditional weather and climate models estimate uncertainty as the spread of an ensemble, constructed by the model inputs perturbations (Grimit & Mass, 2007). The ensemble spread is defined

as a standard deviation of predictions. Previous research (Scher & Messori, 2021) showed that, when using neural networks, ensembles of models with similar architectures (homogeneous) provide similar results. Models weights in the ensemble have to be perturbed with retraining, dropout, etc. Moreover, there are premises that an ensemble of diverse architectures (heterogeneous) might provide better uncertainty estimation (Zaidi et al., 2022).

In our research we construct both homogeneous and heterogeneous ensembles and compare their spread as a predictor for the uncertainty estimation for the model selection mechanism. The suggested pipeline does not impose additional costs as all the models do not need to be modified or retrained.

## RESULTS

### FORECASTING

While designing the experiments, we focused on evaluating the performance and stability of various forecasting models. Our results reveal a trade-off between achieving high computer vision metrics and maintaining forecast stability — while some models excel in certain metrics, their forecasts can be less consistent. However, we found that an ensemble of four high-performing models with diverse architectures — namely MotionRNN, Vid-ODE, UNet, and rUNet — offers robust uncertainty estimation. The most significant improvement over the baseline across nearly all metrics was achieved using a uncertainty-aware model switching scheme which utilized an rUNet backbone, an autoregressive UNet, and DMVFN as a robust alternative (see Figure 3a for schematic representation).

A summary of the metrics evaluated on the test subset for all trained models with uncertainty-aware model switching is presented in Table 2. Figure 4 shows detailed improvements over the baseline, broken down by month and lead time. Figure 13 illustrates the detailed RMSE by date for individual models, along with the corresponding ensemble spreads for several configurations. Examples of predictions are provided in Figure 11. Using the mean of ensembles instead of model selection yields only a marginal improvement in the final metrics, as shown in Table 3.

Table 2: Summary of the test metrics (lower is better) for models with confidence-based mixture with DMVFN as a robust model; the uncertainty is estimated by the ensemble spread of predictions from MotionRNN, Vid-ODE, UNet, and rUNet models. The standard deviation for the best model (rUNet) is estimated by training with three random initializations.

| Model | MSE | 1 - SSIM | 1 - MS-SSIM $(\times 10^{-3})$ | IIEE@15 | IIEE@30 | IIEE@50 $(\times 10^{-2})$ | IIEE@75 |
|---|---|---|---|---|---|---|---|
| Persistence | 11.2 | 9.8 | 5.6 | 11.5 | 10.4 | 11.0 | 7.3 |
| Linear | 9.8 | 9.1 | 5.2 | 14.0 | 9.6 | 11.1 | 7.7 |
| DMVFN | 10.0 | 8.8 | 5.1 | 11.7 | 10.2 | 10.8 | 6.9 |
| IAM4VP | 8.7 | 10.5 | 5.5 | 14.7 | 10.6 | 11.0 | 7.1 |
| Neural ODE | 8.3 | 9.3 | 4.9 | 12.1 | 10.1 | 10.7 | 6.2 |
| MotionRNN | 7.3 | 9.0 | 4.7 | 11.4 | 9.3 | 9.9 | 5.9 |
| Vid-ODE | 7.5 | 8.7 | 4.7 | 12.1 | 9.2 | 9.7 | 5.7 |
| UNet | 7.7 | **8.2** | 4.6 | 12.1 | 9.3 | 9.6 | 6.0 |
| rUNet | **6.8**±0.2 | 8.3±0.2 | **4.5**±0.1 | **10.0**±1.1 | **9.0**±0.3 | **9.2**±0.2 | **5.3**±0.2 |

Table 3: Summary of the test metrics for ensembles. uncertainty-aware model switching to DMVFN is utilized. "rUNet x3" stands for mean forecast of three retrained versions of rUNet. "Best 4" stands for mean prediction from MotionRNN, Vid-ODE, UNet, and rUNet models.

| Ensemble | MSE | 1 - SSIM | 1 - MS-SSIM $(\times 10^{-3})$ | IIEE@15 | IIEE@30 | IIEE@50 $(\times 10^{-2})$ | IIEE@75 |
|---|---|---|---|---|---|---|---|
| rUNet x3 | 6.7 | 8.3 | 4.5 | 10.0 | 8.9 | 9.1 | 5.3 |
| Best 4 | 6.6 | 8.2 | 4.4 | 11.2 | 8.7 | 9.3 | 5.2 |

Following the Grigoryev's work (Grigoryev et al., 2022), models trained to produce 3-day forecasts were also tested with 10-day outputs without any additional fine-tuning. The results are presented

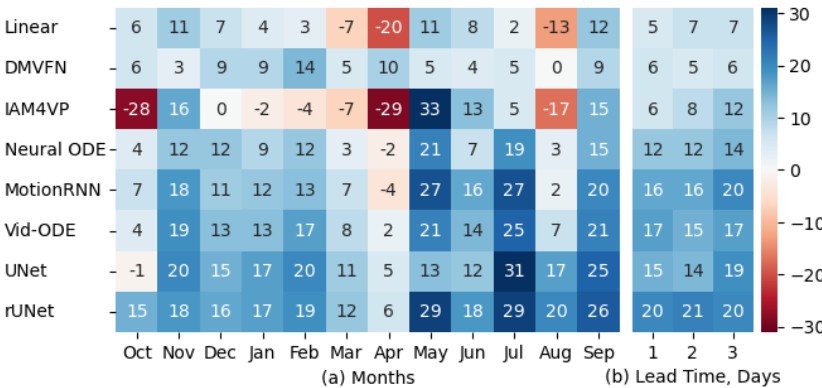

Figure 4: The RMSE percentage improvement over persistence baseline for each month (a), and for each lead time in days (b), over the test subset. The colormap is shared.

in Figure 9. Linear and ODE-based models accumulate errors exponentially, degrading over persistence after the 5-th day. Other models errors increase linearly over time, providing stable improvement over persistence, except IAM4VP which manages to overcome all other models after the 7-th day of the forecast.

Due to the irregular intervals at which satellites capture the target region, a strong correlation between model performance and the length of gaps between valid images is expected. This relationship is illustrated in Figure 10, where RMSE generally increases linearly until the gap length surpasses the models' input size. Once this threshold (7 days) is exceeded, the RMSE approaches that of the persistence baseline.

GAP FILLING

The developed pipeline is particularly useful for filling gaps in SAR images, a common issue in satellite data. Building on the approach proposed by Appel (Appel, 2024), this gap-filling process can be effectively performed as a 1-day forecast. To improve accuracy, we combine forward and backward forecasts, denoted as $y_F$ and $y_B$, respectively. By incorporating the uncertainty estimates of these forecasts, $\sigma_F$ and $\sigma_B$, we can weight them appropriately and calculate an overall uncertainty using a harmonic mean:

$$y = \frac{\sigma_B}{\sigma_F + \sigma_B}y_F + \frac{\sigma_F}{\sigma_F + \sigma_B}y_B, \quad \sigma = \frac{2\sigma_F\sigma_B}{\sigma_F + \sigma_B} \tag{3}$$

A key advantage of this approach is that it does not require retraining the models. We evaluated the performance of this gap-filling method using a leave-one-out cross-validation technique (Kohavi, 1995). For comparison, we also tested the pretrained AdaCoF model (Lee et al., 2020), which is one of the state-of-the-art models for video interpolation. As shown in Table 4, our pipeline achieved a strong $R^2$ value of 87.7%. This is consistent with similar $R^2$ values reported in the literature (Weiss et al., 2014; Appel, 2024) for gap-filling in satellite imagery under similar conditions, such as missing swaths up to 500 km wide and 1 kilometer resolution.

Table 4: Summary of gap filling metrics obtained during a leave-one-out validation. Uncertainty-aware mixture of rUNet and DMVFN is utilized for forward and backward forecasts, where the "Forward+Backward" is a uncertainty-weighted mean. The input channels related to wind and currents are reversed for the backward run. The best metric values in each column are highlighted in bold.

| Model | MSE | 1 - SSIM | 1 - MS-SSIM | IIEE@15 | IIEE@30 | IIEE@50 | IIEE@75 |
| --- | --- | --- | --- | --- | --- | --- | --- |
| | | $(\times 10^{-3})$ | | | $(\times 10^{-2})$ | | |
| AdaCoF | 7.3 | 8.3 | 4.5 | **8.0** | 7.9 | 9.2 | 6.0 |
| Forward | 6.5 | 8.1 | 4.4 | 8.6 | 8.5 | 9.2 | **5.3** |
| Forward+Backward | **6.0** | **7.6** | **4.1** | 9.1 | **7.7** | **8.7** | 5.7 |

## DISCUSSION

This research addresses the critical challenge of short-term regional sea ice forecasting, exploring a variety of approaches to improve prediction accuracy and reliability. Among the methods investigated, modern deep learning models for video prediction were tested for their potential in forecasting sea ice dynamics. However, the performance of these models is constrained by several factors, including the scarcity of high-resolution data, the complex physical processes governing sea ice behavior, the stochastic nature of daily ice dynamics, and the discontinuities present in ice sheet structures. We argue that these domain specialties mostly affect motion-related elements of video prediction models like flow estimation and prediction, see appendix C for further details.

UNet-based models deliver the best individual results, whereas state-of-the-art video prediction models struggle to surpass baseline performance, though they do offer varying levels of stability. It could be argued that the DMVFN model fails to accurately reproduce sea ice thermodynamics due to its architectural limitations, which, paradoxically, contribute to more stable forecasts. On the other hand, IAM4VP, while efficient at modeling different dynamics with minimal computational cost, produces the most unstable predictions, likely due to the lack of sufficient training data.

Advanced use of uncertainty-aware model switching scheme can further enhance the metrics. The ensemble spread of heterogeneous architectures provides accurate uncertainty estimation for the forecasted fields. Although the model-selection mechanism reduces the final spread-error correlation, the total variance in model error can still be explained up to 87% by accounting for the sea ice concentration and its rate of change (see Figure 14).

## CONCLUSIONS

In this research article, we address the challenge of predicting ice conditions in the Gulf of Ob, a region characterized by complex ice formation dynamics influenced by the interaction of saline water and freshwater. We explore the potential of machine learning methods as an alternative to traditional numerical sea ice models, aiming to improve forecasting accuracy and efficiency.

Our key findings reveal that even modern state-of-the-art machine learning models can not achieve sufficient forecasting performance solely. Furthermore, domain-aware data preprocessing and augmentations are essential to train deep learning models for this task. All models struggle due to lack of training data, long gaps in it and complex sea ice dynamics, leading to tricky fidelity-stability trade-off. Although usage of ensembles cannot significantly improve average models performance, it helps to eliminate high errors due to outliers in data, especially in spring season, thus increasing overall system reliability. We consider also an interesting finding that the different ML models capture different aspects of the ice dynamics in such a way that their ensemble gives a reliable forecast uncertainty quantification, as the spread-error correlation coefficient reaches 87%. To overcome the aforementioned trade-off we construct the uncertainty-aware model switching scheme, that provides both stable and explainable forecasts while improving general performance. The mixture of the rUNet and DMVFN architectures provides the best computer vision and geophysical metrics and beats baselines by a wide margin.

Future research directions include developing models that can effectively capture the dynamics of ice formation and melting is crucial. Additionally, addressing the limitations of current approaches through more advanced architectures and techniques can also be beneficial. Further advancements in sea ice forecasting will not only improve maritime navigation safety but also deepen our understanding of complex sea ice dynamics.

## REPRODUCIBILITY

The developed source code was attached to the manuscript as a supplementary material. The preprocessed dataset is available upon written request. The processing procedure and the training of the models is described in the Methods section of the paper. Both code and dataset will be published and available after the end of double-blind review.

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

# APPENDIX

## A   DATA DESCRIPTION

### TARGET REGION

The Gulf of Ob, located at the mouth of the Ob River in the Arctic Ocean, is the world's longest estuary, stretching approximately 1,000 km between the Gyda and Yamal peninsulas (Lapin et al., 2011). It is relatively shallow, with depths averaging 10 to 12 meters, limiting heavy sea transport.

The Taz Estuary, formed by the Taz River, spans about 330 km from Tazovsky to the Gulf of Ob, with an average width of 25 km. It flows north to south and then bends westward before merging with the Gulf of Ob, contributing to one of the largest estuarine systems in the world.

This region is important for sea ice forecasting and research due to its highly variable ice conditions influenced by seasonal changes and river discharge (Osadchiev et al., 2021). It's a sensitive indicator of climate change and has significant economic and strategic value due to its location near major shipping routes and natural resources (Tretiakov & Shiklomanov, 2022). The unique interaction between river outflows and the sea creates distinctive ice patterns, making it a key area for studying sea ice dynamics and improving predictive models(Tikhonov et al., 2022). Additionally, sea ice in this region affects local ecosystems and communities, highlighting the broader impacts of environmental changes on ecology and society.

### INPUT FIELDS

Our neural network model utilizes a number of input channels (fields) that come from three sources: Sentinel-1(Sentinel-1), Global Ocean Physics Reanalysis (GLORYS)(GLO), and historical data from meteostations(Weather & Climate) (see Figure 5 for detailed information). Sentinel-1 SAR images are interpolated conservatively to match the input resolution (1 km), GLORYS fields are interpolated bilinearly, data from meteostations is interpolated between discrete points (where the meteostations are located) using RBF interpolation method with thin plate splines (Wahba, 1990). The details on resulting channels and preprocessing for input data are described in Table 5.

Table 5: Description of input channels. GLORYS channels are interpolated bilinearly. Meteodata is interpolated using radial basis function interpolation.

| Source | Scale | Channel | Normalization |
|---|---|---|---|
| Sentinel-1 | 1 km | SAR HV | $U(0,1)$ |
| | | SAR HH | $U(0,1)$ |
| GLORYS | 5 km | Bottom Temperature | $U(-1,1)$ |
| | | Mixed Layer Thickness | $U(-1,1)$ |
| | | Surface Salinity | $U(-1,1)$ |
| | | Surface Temperature | $U(-1,1)$ |
| | | Sea Ice Velocity (u) | $N(0,1)$ |
| | | Sea Ice Velocity (v) | $N(0,1)$ |
| | | Sea Height | $N(0,1)$ |
| Meteostations | – | Relative Humidity | $U(0,1)$ |
| | | Air Pressure | $N(0,1)$ |
| | | Air Temperature | $N(0,1)$ |
| | | Wind Velocity (u) | $N(0,1)$ |
| | | Wind Velocity (v) | $N(0,1)$ |

| Name | Lat | Lon |
|------|-----|-----|
| New Port | 67.68 | 72.88 |
| Seyakha | 70.17 | 72.52 |
| Tambay | 71.48 | 71.82 |
| Popov Island | 73.33 | 70.05 |
| Nyda | 66.63 | 72.93 |
| Yar-Sale | 66.80 | 70.83 |
| Beloyarsk | 66.87 | 68.13 |
| Laborovaya | 67.64 | 67.56 |
| Poluy | 66.03 | 68.68 |
| Nadym | 65.47 | 72.67 |
| Tazovsky | 67.47 | 78.73 |
| Antipayuta | 69.08 | 76.85 |
| New Urengoy | 66.10 | 76.78 |
| Urengoy | 65.95 | 78.40 |
| Vilkitsky Island | 73.50 | 76.00 |
| Dixon Island | 73.52 | 80.40 |
| Marresal | 69.71 | 66.80 |
| Ust-Kara | 69.25 | 64.93 |
| Karaul | 70.08 | 83.17 |
| Sopochnaya Karga | 71.87 | 82.70 |
| Izvestia Islands CEC | 75.95 | 82.95 |
| Cape Zhelaniya | 76.95 | 68.55 |
| Gyda | 70.88 | 78.47 |

(a)  (b)

Figure 5: The map (a) and coordinates (b) of meteorological stations used, along with the target region outlined in red. Available sea surface area is 120,559 km$^2$. The area of interest is 51,262 km$^2$.

SAR ESTIMATES SEA-ICE CONDITIONS

In comparison to other potential target variables, such as GLORYS reanalysis, which lacks quality in the Gulf and which is mostly uncorrelated with other sources (see Figure 7), GLORYS operative analysis and forecasts, which lack historical records essential for data-driven approaches, and AMSR (Ludwig et al., 2020), which is partly dependent on cloud conditions and seasons, Sentinel-1 SAR imagery emerges as a superior choice for high-resolution sea ice forecasting models.

While the direct comparison between SAR and calculated sea ice concentrations is not strictly fair, the techniques of retrieval and mapping sea ice conditions from SAR imagery are well-known. Sentinel-1 C band consists of four polarizations, for the purpose of forecasting ice conditions we utilize colorized HV polarization (Sentinel Hub). Figure 6 shows the comparison of monthly-averaged sea ice concentrations from several sources.

## B  METRICS

In our research we utilize two type of metrics. First, we use the common computer vision ones: the mean squared error (MSE) also known as L2-distance; the structural similarity index measure (SSIM), a metric used to assess the human-perceived quality of digital images and videos (Wang et al., 2004), predominantly used in computer vision; and the multi-scale structural similarity index measure (MS-SSIM), which extends the concept of the SSIM by evaluating image quality at various

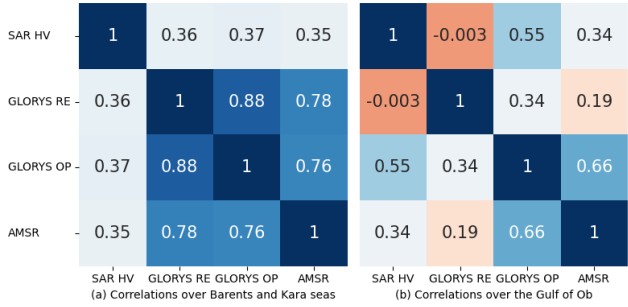

Figure 6: Mean cell-wise sea ice concentration correlation between several data sources: Sentinel-1 SAR (Sentinel-1), GLORYS operative and reanalysis (GLO), and AMSR (Ludwig et al., 2020)

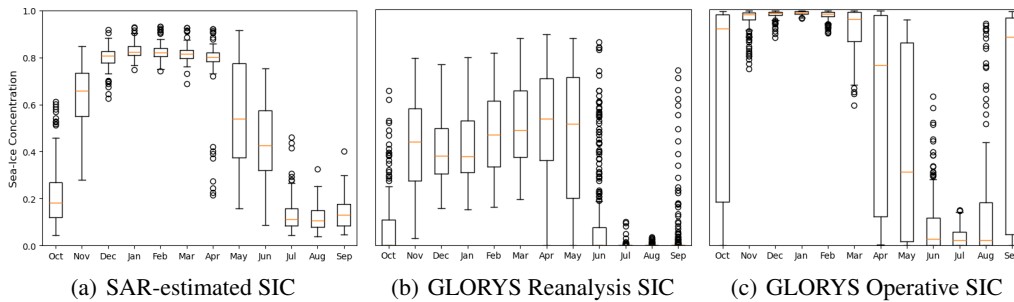

Figure 7: Box and whisker plots of SIC data distribution in the region in GLORYS and Estimated from SAR-images for different months from all available range of time, aggregated over target region. The box extends from the 25th percentile to the 75th percentile; whiskers extend the box by 1.5x of its length. The orange line is the median (50th percentile); SAR-estimated SIC is a normalized mean absolute value of SAR signal with dropped frames with more than 50% missing values.

scales (Wang et al., 2003). The MS-SSIM approach uses the fact that the human eye perceives picture quality differently across varying resolutions, making it a more comprehensive metric for assessing the perceived quality of digital images and videos. Second, we use a geophysical metric specific for sea ice condition analysis and forecast: the integrated ice edge error at level $c$ (IIEE at c), a metric of similarity between ice sheets, where ice edges are chosen at the certain level of concentration $c$ measured in percents (Goessling et al., 2016):

$$IIEE@c = \frac{1}{n} \sum_{i=1}^{n} \frac{1}{S} \sum_{h,w} [(y_i > c) \neq (\hat{y}_i > c)]\, dS_{hw}, \quad S = \sum_{h,w} dS_{hw} \qquad (4)$$

where $y_i$ and $\hat{y}_i$ are linearly normalized into range $[0, 100]$. Usually parameter $c$ is set to 15%, however we can not assume a linear relationship between ice concentration and SAR images, thus we will exploit several values for $c$.

## C  OPTICAL FLOW ESTIMATION FOR SEA ICE

We argue that a fundamental challenge with modern machine learning models is their inability to replicate the complex mechanics of sea ice in coastal regions. The poor performance in capturing fine-scale ice mechanics is not unique to any one method but is a common issue across various approaches at the target resolution. For instance, neither modern sea ice motion vectors (GLO; Noriaki et al., 2013) nor optical flow estimation methods (Farnebäck, 2003; Weinzaepfel et al., 2023; Sun et al., 2018) are well-suited for high-resolution ice velocity estimation. This degradation in quality when transitioning to higher resolutions is illustrated in Figure 8. Moreover, deep learning methods for optical flow estimation may be overfitted on common images and lack the generalization needed for sea ice SAR imagery. Consequently, motion information is scarcely useful for predictions in the region of interest.

Low quality of optical flows might be caused by high homogenity of ice-sheet surface and stochastic local dynamics on 1-day scale. For similar reasons one can expect state-of-the-art models on video-prediction task to fail on ice-dynamics forecasting, as their architectures sometimes are based on optical flow estimation and prediction, and they assume the simple mechanical and deterministic dynamics.

| Resolution: | 1 km | 2 km | 4 km | 8 km | 16 km | 32 km |
|---|---|---|---|---|---|---|
| Persistence | 7.0 | 7.2 | 7.2 | 7.4 | 7.1 | 7.5 |
| Glorys Operative | 8.6 | 8.8 | 8.2 | 8.4 | 6.9 | **5.6** |
| AMSR JAXA SIM_R | 7.1 | 6.9 | 6.6 | **6.4** | **6.3** | 5.8 |
| Farneback | **6.7** | **6.6** | **6.4** | **6.4** | 6.5 | 5.9 |
| CrocoFlow | 6.8 | 6.7 | 6.5 | **6.4** | 6.5 | 5.9 |
| PWC-Net | 6.9 | 6.8 | 6.5 | **6.4** | 7.0 | 8.2 |

Figure 8: Mean Squared Error (MSE) ($\times 10^{-3}$) between next-day images and previous-day images, warped using estimated flow from following sources: GLORYS Operative model (25 km resolution), AMSR JAXA SIM-R (50 km resolution), and several Optical Flow models, such as the algorithmic Farneback method and state-of-the-art neural networks. The best MSE values for each resolution are highlighted in bold.

## D  ABLATION STUDY

This section contains ablation studies for crucial parts of training and prediction pipelines: filtration of SAR-imagery artifacts (Table 6), proper augmentations to leverage unbalance and lack of data (Table 7), and the usage of confidence-based model selection and ensembles (Tables 8, 9).

Ensembles usually provide minor improvements except for IIEE@15 metric. However, confidence based model selection suppresses the advantages of ensembles. The usage of model selection (depicted at Table 2) increases MSE and IIEE@75 by 12%.

Table 6: Summary of the metrics obtained by testing individual models without data preprocessing. Raw data have high noise-to-signal ratio due to thermal artifacts. These artifacts simultaneously provide huge bias in metrics and corrupt loss function making the models learn filtration and smoothing rather than forecasting sea ice dynamics.

| Model | MSE | 1 - SSIM $(\times 10^{-3})$ | 1 - MS-SSIM | IIEE@15 | IIEE@30 $(\times 10^{-2})$ | IIEE@50 | IIEE@75 |
|---|---|---|---|---|---|---|---|
| Persistence | 18.5 | 9.6 | 6.8 | 17.3 | 12.5 | 10.2 | 8.5 |
| Linear | 15.7 | 8.9 | 6.2 | 17.6 | 12.9 | 10.0 | 8.4 |
| DMVFN | 16.7 | 8.5 | 6.2 | 17.2 | 12.4 | 9.9 | 8.1 |
| IAM4VP | 16.8 | 9.7 | 6.6 | 18.5 | 15.5 | 11.5 | 10.4 |
| Neural ODE | 13.7 | 8.5 | 5.8 | 17.0 | 12.4 | 9.9 | 7.8 |
| MotionRNN | 12.7 | 8.1 | 5.4 | 16.2 | 11.9 | 9.2 | 7.6 |
| Vid-ODE | 12.2 | 7.8 | 5.4 | 16.5 | 11.4 | 8.7 | 7.1 |
| UNet | 13.0 | 7.6 | 5.4 | 15.7 | 11.4 | 9.1 | 7.4 |
| rUNet | 13.6 | 7.8 | 5.5 | 15.7 | 12.0 | 9.3 | 7.6 |

Table 7: Ablation studies for the augmentations for the best performing model (rUNet). Geometric augmentations are shifts and rotations (treating input as an image). The physical augmentations are modifications of geometrical ones with corresponding transform (rotations and flips) of physical fields (currents and winds). "Proposed" states for superposition of Physical and Frameout augmentations.

| Augmentation | MSE | 1 - SSIM $(\times 10^{-3})$ | 1 - MS-SSIM | IIEE@15 | IIEE@30 $(\times 10^{-2})$ | IIEE@50 | IIEE@75 |
|---|---|---|---|---|---|---|---|
| None | 8.9 | 9.2 | 4.9 | 11.9 | 9.4 | 10.8 | 7.0 |
| Geometric | 8.7 | 9.0 | 4.8 | 12.9 | 10.4 | 10.9 | 6.6 |
| Physical | 7.8 | 8.4 | 4.6 | 10.1 | 9.1 | 10.0 | 6.1 |
| Proposed | 7.6 | 8.3 | 4.6 | 10.0 | 9.0 | 9.8 | 6.0 |

Table 8: Summary of test metrics for individual models with proposed preprocessing and augmentations.

| Model | MSE | 1 - SSIM | 1 - MS-SSIM | IIEE@15 | IIEE@30 | IIEE@50 | IIEE@75 |
|---|---|---|---|---|---|---|---|
| | | $(\times 10^{-3})$ | | | $(\times 10^{-2})$ | | |
| Persistence | 11.2 | 9.8 | 5.6 | 11.5 | 10.4 | 11.0 | 7.3 |
| Linear | 9.9 | 9.1 | 5.2 | 14.2 | 9.6 | 11.0 | 8.0 |
| DMVFN | 10.0 | 8.8 | 5.1 | 11.7 | 10.2 | 10.8 | 6.9 |
| IAM4VP | 9.5 | 10.6 | 5.6 | 14.7 | 10.6 | 11.4 | 8.1 |
| Neural ODE | 8.6 | 9.3 | 4.9 | 12.1 | 10.1 | 10.7 | 6.8 |
| MotionRNN | 8.0 | 9.0 | 4.7 | 11.4 | 9.3 | 10.3 | 6.5 |
| Vid-ODE | 7.7 | 8.6 | 4.7 | 12.2 | 9.2 | 9.6 | 6.0 |
| UNet | 8.3 | 8.2 | 4.6 | 12.1 | 9.5 | 9.9 | 6.5 |
| rUNet | 7.6 | 8.3 | 4.6 | 10.0 | 9.0 | 9.8 | 6.0 |

Table 9: Summary of test metrics for ensembles. "rUNet x3" stands for mean forecast of 3 retrained versions of rUNet. "Best 4" stands for mean of MotionRNN, Vid-ODE, UNet, and rUNet predictions.

| Ensemble | MSE | 1 - SSIM | 1 - MS-SSIM | IIEE@15 | IIEE@30 | IIEE@50 | IIEE@75 |
|---|---|---|---|---|---|---|---|
| | | $(\times 10^{-3})$ | | | $(\times 10^{-2})$ | | |
| rUNet x3 | 7.1 | 8.3 | 4.5 | 11.2 | 8.8 | 9.2 | 6.1 |
| Best 4 | 7.1 | 8.2 | 4.4 | 11.2 | 8.7 | 9.4 | 6.1 |

# E SUPPLEMENTARY FIGURES

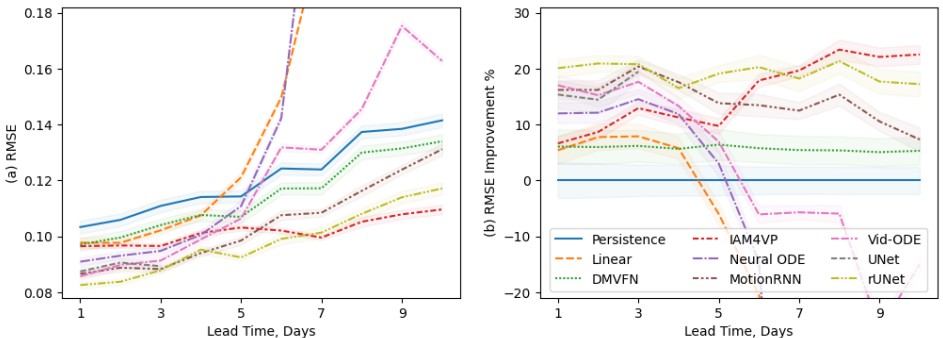

Figure 9: (a) RMSE and (b) its percentage improvement over persistence baseline for each extended lead time in days over the test subset.

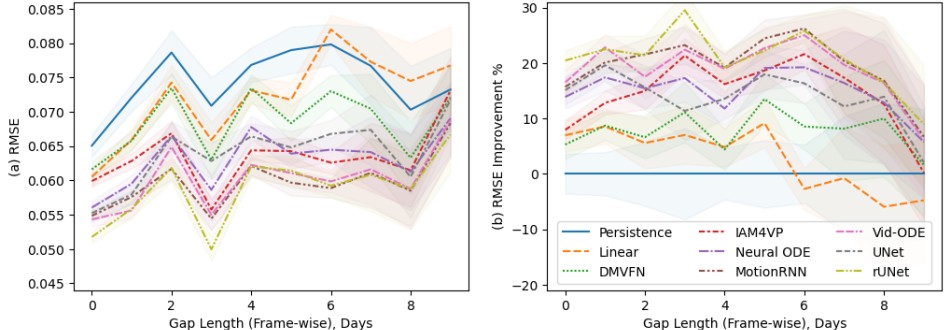

Figure 10: (a) RMSE and (b) its percentage improvement over persistence baseline in dependence of preceding SAR gap length.

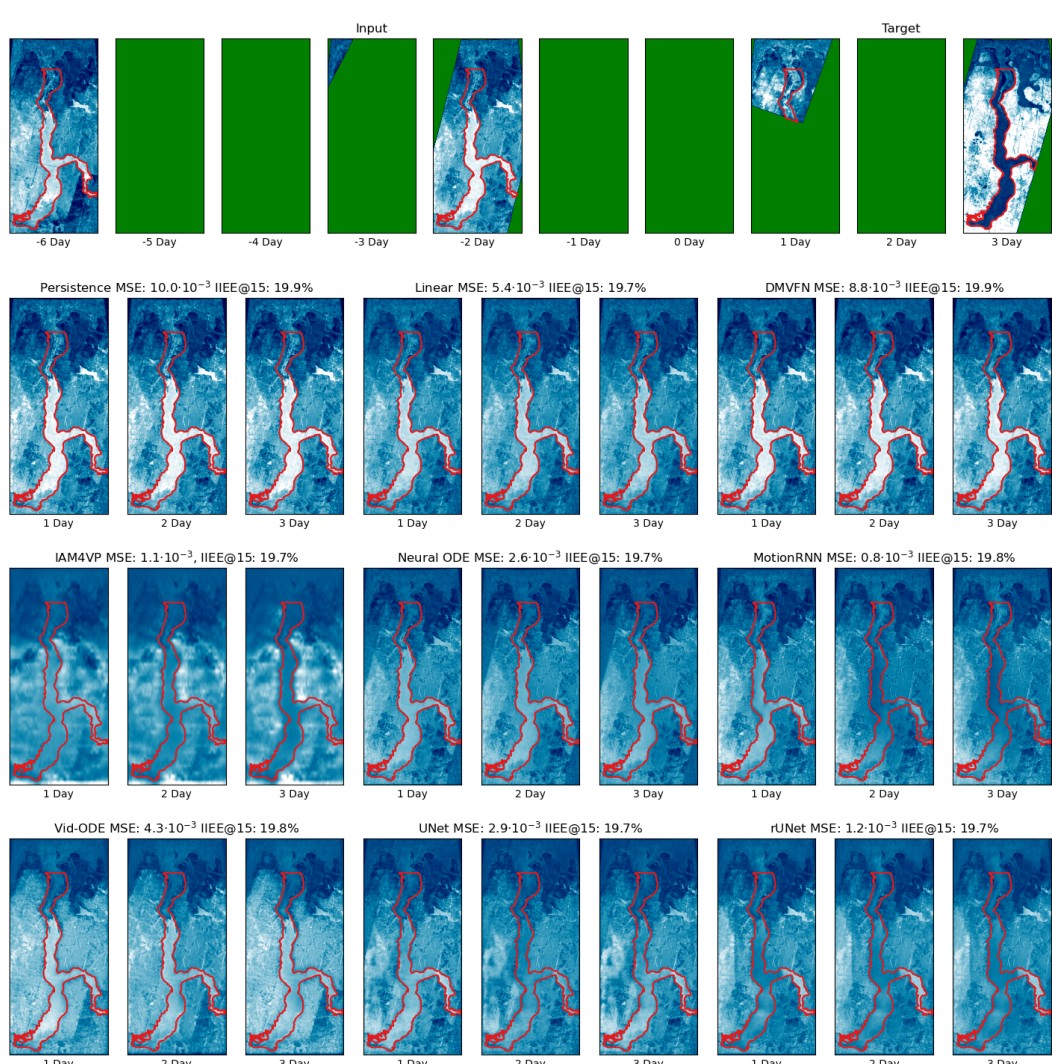

Figure 11: The example of forecasts. Timestamps represent shifts from the 25-05-2023. The target region is outlined with a red line. The missed data in an input and a target sequences is represented by green color.

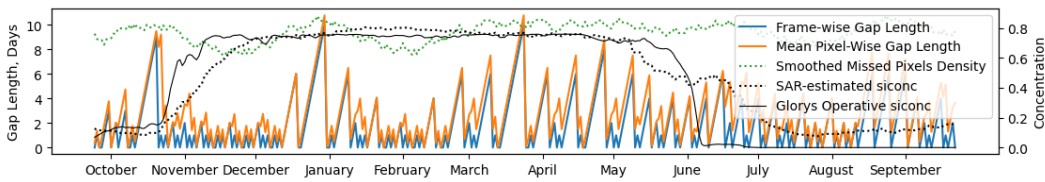

Figure 12: Distance to the nearest valid data frame-wise (blue) and mean value pixel-wise (orange); concentration of missing values smoothed with half-month-wide rolling window; operative glorys sea-ice concentration; and mean SAR-value as an estimation of sea-ice concentration. All curves are evaluated over test subset.

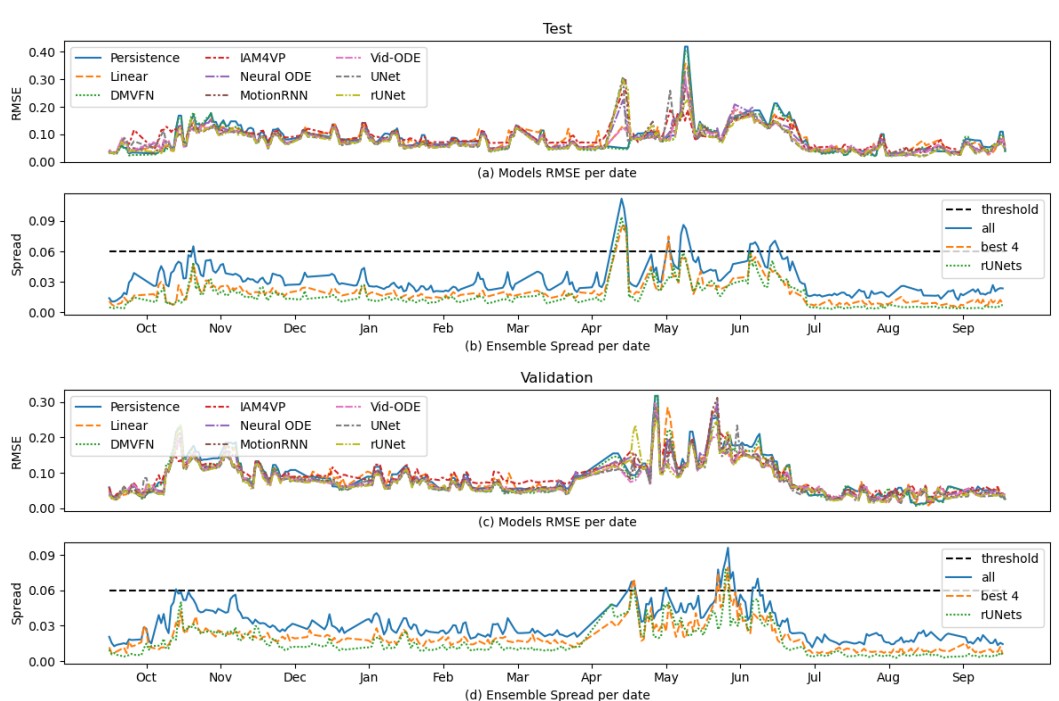

Figure 13: RMSE timelines for all the models over test (a) and validation (c) subsets, ensemble spreads over test (b) and validation (d) subsets, where 'best 4' stands for MotionRNN, Vid-ODE, UNet, and rUNet, 'rUNets' stands for 3 different initializations of rUNet model. Threshold is tuned on validation subset for consequent use in confidence-based model selection during testing.

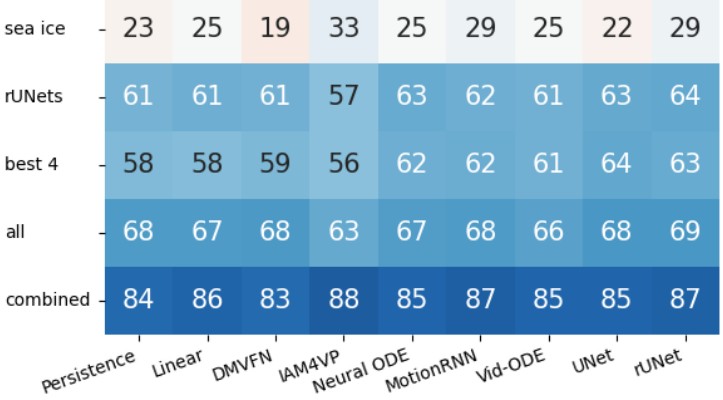

Figure 14: Correlation (in percents) between models RMSE (with confidence-based model selection) and several features: sea ice concentration, ensemble spread, and their learned linear combination.

