# OpenReview forum: "Data-Driven Uncertainty-Aware Forecasting of Sea Ice Conditions in the Gulf of Ob Based on Satellite Radar Imagery"
_ICLR.cc/2025/Conference — Submitted to ICLR 2025_

### Official Review · Reviewer_CM4n · 2024-10-23

**Soundness:** 3
**Presentation:** 3
**Contribution:** 2
**Rating:** 5
**Confidence:** 4

**Summary:**

The paper presents a data-driven approach for forecasting sea ice conditions in the Gulf of Ob by leveraging advanced video prediction models originally developed for computer vision tasks. The authors utilize sequences of radar images from Sentinel-1, weather observations, and GLORYS forecasts to predict future sea ice conditions. They address challenges related to data irregularity and missing values through domain-specific preprocessing and augmentation techniques. The paper also introduces an ensemble-based approach for uncertainty quantification and proposes a confidence-based model selection scheme to enhance forecast accuracy and robustness.

While the paper tackles a relevant and practical problem, it primarily applies existing deep learning models to a new domain without significant methodological innovations. The contributions are more engineering-focused, adapting existing models for sea ice forecasting without introducing new algorithms or theoretical advancements. The improvements over baseline models are modest, and there is limited discussion on the practical significance of these improvements or how they translate to real-world applications.

**Strengths:**

Application of Deep Learning to Sea Ice Forecasting: The paper addresses a relevant and practical problem by applying advanced video prediction models to sea ice forecasting in the Gulf of Ob. This cross-disciplinary application showcases the potential of deep learning in geophysical tasks.

Data Preprocessing Techniques: The authors develop domain-specific data preprocessing and augmentation methods to handle the challenges of Arctic satellite imagery, such as data irregularity and missing values. This is crucial for improving model performance on imperfect real-world data.

Uncertainty Quantification: Introducing an ensemble-based approach for uncertainty estimation and a confidence-based model selection scheme adds value by enhancing forecast robustness and providing a mechanism to assess prediction reliability.

**Weaknesses:**

* Lack of Novel Methodological Contributions: The paper primarily applies existing video prediction models to a new dataset without significant modifications or novel methodological developments. This limits its contribution to the advancement of machine learning techniques.

* Engineering Focus Over Research Innovation: The work focuses more on engineering implementation and practical adaptation rather than introducing new theoretical insights or advancements in machine learning.

* Modest Improvements Over Baselines: The improvements over baseline models are modest. The paper lacks a deep analysis of the practical significance of these improvements, especially in operational contexts.

* Insufficient Theoretical Analysis: There is a lack of in-depth theoretical analysis or exploration of why certain models perform better in this context, which could provide valuable insights to the research community.

**Questions:**

* Novelty of Contributions: Can the authors clarify what novel methodological contributions are presented beyond applying existing models to a new dataset? Are there any new algorithms, architectures, or theoretical insights introduced?

* Model Adaptations: Did the authors make any significant adaptations or improvements to the video prediction models to better suit sea ice forecasting, or were the models used off-the-shelf?

* Evaluation of Practical Significance: How do the modest improvements over baselines translate to practical benefits in operational forecasting? Are these improvements significant enough to impact real-world applications?

* Generalizability: Can the authors discuss the potential generalizability of their approach to other regions or types of geophysical forecasting? What are the limitations?

---

> ### Author Response · Authors · 2024-11-26
>
> We thank the reviewer for their constructive feedback. We appreciate the acknowledgment of the novelty of the task and the importance of the introduced augmentations. Below, we address the specific concerns and questions raised in the review:
>
> ### Response to weaknesses:
> 1. **Lack of Novel Methodological Contributions.** We consider our main methodological contribution the proposed uncertainty quantification and uncertainty aware model switching scheme leveraging different ML models’ strengths and flaws. In particular we have shown that the ensemble of different ML methods produces a good uncertainty estimate for the complex geophysical task of sattelite radar image forecasting and the use of this estimate can improve the quality of the resulting prediction by switching between the models. We have improved the Contributions section to reflect this better. As a secondary contribution, we developed augmentations that combine physical and geometrical transformations to better align with geophysical characteristics and include handling irregularly missing values due to satellite coverage limitations.
> 2. **Engineering Focus Over Research Innovation.**
> We introduce data filtration and task-specific augmentations, which combine physical and geometrical transformations, and leverages the problem of irregularly missing values due to limitation of satellite coverage; we develop the uncertainty-aware pipeline, which results in performance improvement and provide uncertainty quantification, and proof the efficacy of heterogeneous ensemble in geophysical applications. We argue that our work enhances the connection between video prediction and the challenging task of high-resolution sea ice forecasting, which alignes well with ICLR relevant topics, such as applications of DL in geophysical and climate sciences.
> 3. **Improvements Over Baselines.** The improvements we achieved (1.5-2x better than baselines) are significant in the context of Arctic sea ice forecasting, where even small gains translate to operational benefits. The uncertainty estimation, being an additional result of the proposed approach, is also useful for the practical tasks such as icebreaker caravan route planning with risks consideration
> We also included an ablation study to explore the contribution of various techniques, such as uncertainty modeling and specialized augmentations, which helped achieve these improvements. Also Figure 6 in Appendix provides a brief analyis, showing that modern NWPs, like that producing GLORYS product struggle to forecast sea ice conditions in the Golf of Ob as well.
> 4. **Theoretical Analysis.** We acknowledge the importance of theoretical analysis and have provided additional context in the revised manuscript. The performance of optical flow estimation and prediction, an essential architecture feature of video prediction models, is limited in low-variability ice sheet regions and due to complexity of stochastic physics (see Appendix C). These insights highlight why video prediction models struggle in this domain, providing a foundation for future research. While a deeper theoretical analysis of model behavior is valuable, we note that this falls outside the primary scope of this work, which is centered on applying and adapting ML models for a novel and challenging task.

---

> > ### Comment · Reviewer_CM4n · 2024-11-27
> >
> > > Lack of Novel Methodological Contributions:
> >
> > * The authors argue that their primary contribution is the uncertainty-aware model switching scheme and the use of heterogeneous ensembles for uncertainty quantification.
> >
> > * While this contribution is relevant and tailored to geophysical applications, it is incremental rather than groundbreaking in machine learning research. The augmentations and handling of missing data are secondary, domain-specific engineering
> > solutions.
> >
> > * The methodological contributions do not push the boundaries of ML research; they apply existing techniques to a new domain, which aligns more with engineering than novel innovation.
> >
> > > Engineering Focus Over Research Innovation:
> >
> > * The authors justify their engineering focus by framing it as necessary for bridging video prediction and geophysics, which aligns with ICLR topics. They emphasize augmentations, the pipeline, and heterogeneous ensembles as key contributions.
> >
> > * While this argument is valid for an application-focused paper, it does not elevate the work to the level of innovation expected in a top-tier ML conference. The response highlights practical relevance but fails to address the lack of deeper theoretical advancements.
> >
> > > Improvements Over Baselines:
> >
> > * While speculative about real-world impacts, their explanation and added study provide sufficient evidence of improvement, addressing this concern.
> >
> > > Theoretical Analysis:
> >
> > * The authors include new discussions on why optical flow models struggle in geophysical tasks and provide additional insights in Appendix C. However, they argue that deeper theoretical analysis is beyond the scope of this paper.
> >
> > * The added context is helpful but does not address the lack of rigorous theoretical contributions, which limits the broader significance of their findings in the ML community.

---

> ### Author Response · Authors · 2024-11-26
>
> ### Response to questions:
> 1. **Novelty of Contributions.** Our contributions include: the development of task-specific augmentations combining physical and geometric transformations to address missing data issues in satellite imagery; implementation of an uncertainty-aware pipeline using heterogeneous ensembles, demonstrating their effectiveness in geophysical contexts.
> 2. **Model Adaptations.** Models were adapted to handle missing values, a common issue in satellite-derived data (lines 341-343 and Figure 3 in the manuscript). All models were extended to work with multiple geophysical data channels, departing from the traditional RGB-focused implementations.
> 3. **Evaluation of Practical Significance.** It is challenging to assess the direct practical benefits of these forecasts without conducting a thorough practical study and onboarding the proposed methods for operational use by captains. However, this requires substantial additional work, which we believe lies beyond the scope of this paper. Initially we have consulted the icebreaker first mate who said that the satellite image direct operational forecasts could be of big help for route planning and that’s one of the reasons why we have made this study.
> 4. **Generalizability.** Our approach is expected to generalize well to regions of other Arctic rivers, however, we acknowledge that numerical models may be superior in regions with free sea surfaces. The challenges in sea ice modeling, such as limited satellite coverage and the dynamic nature of ice sheets, make this task uniquely demanding. Also, it would be interesting to apply the approach to glacier movement analysis and forecast. The explored techniques can be helpful in a broad range of geophysical applications, such as ocean biochemistry, ocean plastic pollution, and atmospheric pollution modelling.

---

> > ### Comment · Reviewer_CM4n · 2024-11-27
> >
> > * Novelty of Contributions: Partially addressed my concerns, reiterating the same contributions without significantly enhancing their novelty.
> >
> > * Model Adaptations: Sufficiently addressed my concerns, showing thoughtful adaptations for geophysical challenges.
> >
> > * Evaluation of Practical Significance: Partially addressed my concerns with anecdotal evidence but lacks concrete practical evaluation.
> >
> > * Generalizability: Addressed with a fair discussion of broader applicability.

---

> > > ### Comment · Reviewer_CM4n · 2024-11-27
> > >
> > > The authors have addressed some weaknesses, improving the paper’s clarity and focus. While contributions remain incremental, I just changed my score to "marginally below the acceptance threshold."

---

### Official Review · Reviewer_GA94 · 2024-11-03

**Soundness:** 2
**Presentation:** 1
**Contribution:** 2
**Rating:** 3
**Confidence:** 3

**Summary:**

Short-term forecasts of satellite images at 1km resolution conditioned on past satellite imagery and past meteorological conditions with deep neural network architectures commonly used in video prediction. The networks beat a simple baseline (persistence), but most video prediction methods do not improve over a UNet. Moreover, the presented models struggle due to the inherent sparsity of the satellite time series, yet a new augmentation method (joint geometric transformations of meteo fields and satellite images) is introduced which improves sample efficiency in this data sparse setting.

**Strengths:**

1) The task of satellite-based sea ice forecasting conditioned on meteorology is interesting and sufficiently novel
2) The paper introduces an augmentation strategy which improves performance significantly
3) The work compares many different neural network architectures and includes two simple baselines
4) A domain-specific evaluation metric is used, the Integrated Ice Edge Error.

**Weaknesses:**

1) The results are not convincing. This work trains very large deep neural networks (some over 30Mio parameters) on a very small dataset (training has only ~2200 days). The trained models beat a simple persistence baseline by some margin, but it is unclear what this means, as there is no comparison to any baseline sea ice forecast and there is almost no qualitative evidence presented in this paper. The only qualitative results are shown in Fig. 6, but those are not convincing, there, all models fail to provide a forecast that is somewhat close to the reality at day 3. My impression after reading is that the gaps in the time series and the low availability of past data make the task extremely challenging, such that the models mainly learn a blurred regression to the mean, which in MSE beats persistence.
2) The writing lacks clarity. Many important technical details are not explained well (or not at all), instead the paper is full of fill-words and meaningless phrases that sound like output from a LLM. I'll provide more specific feedback below.
3) It is hard to assess what the contribution of this work is. I see the main novelty in the augmentation strategy, but that is a bit too little for an ICLR paper.
4) The paper emphasizes that fancy video prediction architectures do not outperform an out-of-the-box UNet for satellite image time series forecasting, but instead domain-specific preprocessing is more important. However, this finding is not new, see e.g. Benson et al 2024 https://openaccess.thecvf.com/content/CVPR2024/html/Benson_Multi-modal_Learning_for_Geospatial_Vegetation_Forecasting_CVPR_2024_paper.html - which focusses on vegetation greenness forecasting, but else is very similar in design.
5) Missed opportunity: the work only uses Sentinel 1 at 1km resolution, however the big benefit of the satellite is its high spatial resolution (up to ~15m). At coarser resolution, i doubt Sentinel 1 is the best product, especially due to its temporal sparsity (only ~5-daily). Moreover, the work only uses past meteorological data. Yet, future sea ice motion is likely depending a lot on future weather, hence it would make a lot of sense to include future weather. Ideally, to emulate operational conditions, this would be from stored weather forecasts, but for showing the predictive skill of the map weather -> sea ice, it would also suffice to just use future reanalyis, mentioning a potential further performance degradation at inference time due to the usage of actual forecasts.
6) The evaluation of ensembles is a bit weak. If you provide ensemble forecasts for uncertainty quantification, as a user, i'd most importantly like to see how well they are calibrated, i.e. the skill score. There are further probabilistic metrics like CRPS that should also be looked at. And not just MSE of the ensemble mean.
7) Many formulations in the paper are debatable:  l. 013ff I'd argue the causality is wrong in this sentence. Short-term sea ice forecast are important because they are useful for boats navigating through the arctic sea, not because of global warming and subsequent sea ice loss. ; l. 100ff by comparing the accuracy of model predictions we do not ensure that these predictions contain more than just general trends (what are those anyway?) and we also do not ensure that they contain spatial structures. ; l. 148ff The main reason for the data gaps is that Sentinel 1 is on an orbit that only has a revisit time of 12 days. For some time (until 2021), there were two satellites, which, together with considering off-nadir imagery allowed for an effective revisit time of 2-3 days, now it is 5-6 days. All other factors are minor compared to this one. ; l. 214 I am unaware of any weather capability of Sentinel 1 (what is that anyway?) - however, it may be worth to mention that contrary to passive optical imagery like Sentinel 2, the active sensor of Sentinel 1 can measure surface conditions even if there is cloud cover. ; L. 235 Sentinel 1 has up to ~15m resolution. L. 236 It is only partially true that there are large amounts of historical data: while the size in terms of storage is surely in the petabytes, we have only a very limited (!) historical record of Sentinel 1, just 9 years since 2015.
8) Limited related works section. Only googling once gave me already a very related paper Palerme et al 2024 https://tc.copernicus.org/articles/18/2161/2024/ doing short-term sea ice forecasting with deep learning. The related works section needs to include such works, and ideally you compare the performance of your models to those published in these related works. Furthermore, there is a large stream of literature on satellite image time series forecasting, which seems extremely relevant, but the related works section also misses.

**Questions:**

1) How do you split the different samples in the training period? Do you include forecasts starting at every day? Or only every 10 days to avoid data leakage (one samples target being in another samples input)? --> Following from this, what is your exact sample size for train, val, test: before and after augmentation?
2) How do you do the backward forecast (l. 462)? Are you considering that atmospheric dynamics are not time-reversible due to the second law of thermodynamics?
3) Are you using the same augmentation strategy for all models?
4) Which Sentinel 1 product are you using? How has it been processed? Is it radiometrically terrain corrected?
5) How are you computing the IIEE?
6) Could you explain the filtration in other words again? L.292ff - I did not understand from reading the manuscript.
7) Why the loss MSE - 0.2 SSIM?
8) How do you feed missing inputs to your models?
9) Do you have any idea why the missing values (Fig 2a) were a lot lower during 2016 & 2017? To me it makes little sense and I would rather expect a drop in 2021, when the Sentinel 1B satellite went out of functioning.
10) Have you compared to a climatology? For satellite imagery this seems a very important baseline, see again e.g. Benson et al 2024 https://openaccess.thecvf.com/content/CVPR2024/html/Benson_Multi-modal_Learning_for_Geospatial_Vegetation_Forecasting_CVPR_2024_paper.html
11) I do not understand how the confidence-based mixture with DMVFN (l. 433f) plays a role in the predictions of the models presented in Table 3, can you elaborate?

---

> ### Author Response · Authors · 2024-11-25
>
> We thank the reviewer for their time and constructive feedback. We appreciate the acknowledgment of the novelty of the task and the importance of the introduced augmentations. Below, we address the specific concerns and questions raised in the review:
>
> ### Response to weaknesses:
> 1. **Overparameterization and performance.** We agree on that the task is challenging. However, it is so for numerical models as well, as we point out in the paper. There are two reasons to choose large NN models: firstly, high resolution effectively increases amount of data (one can split the image into patches) and necessitates the usage of model with perception window large enough to capture global dynamics; что secondly, overparameterized models are widely used nowadays because of the specific properties giving them ability to converge to good solutions (probably suboptimal) as demonstrated in many works (e.g. https://www.sciencedirect.com/science/article/pii/S106352032100110X).
> 2. **Text clarity.** We updated the manuscript to improve clarity.
> 3. **Contributions.** We think that our main contribution is also the developed uncertainty-aware pipeline, which results in performance improvement and provides uncertainty quantification, and proof the efficacy of heterogeneous ensemble in geophysical applications. We argue that our work enhances the connection between video prediction and the challenging task of high-resolution sea ice forecasting, which aligns well with ICLR defined scopes.
> 4. **Novelty.** We argue that the task of vegetation forecasting is simpler due to absence of movement, the sea ice dynamics is much more complex. Moreover, our study incorporates the combination of physical and geometrical augmentations and develops the uncertainty-aware pipeline based on UNet, modified for autoregressive forecast and handling missing values (see Figure 3), DMVFN and heterogeneous ensemble.
> 5. **Forecast resolution.** Thanks for a good point and interesting proposal. We selected this resolution because it is at the frontier of modern research on sea ice forecasting and highlights the techniques that would be essential for the increase of resolution in further studies. The 1 km resolution is just enough for planning the icebraker caravan routes in the challenging conditions of the Gulf of Ob which we consider as one of the possible practical uses of our results.
> 6. **Ensemble evaluation.** We study the spread-error correlation of ensembles (see Figure 13). Up to 87% of variance is explained, which proves the heterogeneous ensemble to be efficient in geophysical application. We reckon that further investigation is out of scope of the study.
> 7. **Formulations clarity.** Thank you for pointing out these problematic moments! We fixed the aforementioned comments in the updated manuscript.
> 8. **Related works.** We consciously reduced the scope of related works. We firstly refer to one of the first data-driven sea ice forecasting models, then discuss most relevant modern models, specifically for short-term forecast in one kilometer resolution, and finally discuss works related to uncertainty quantification and gap filling. The study by Palermo et al is focused on 12.5km resolution and hence is out of scope.

---

> ### Author Response · Authors · 2024-11-25
>
> ### Response to questions:
> 1. **How do you split the different samples in the training period?** We include sequences started at every date, so most of the images are inputs in some sequences and targets in others. The models never see the test subset, so we consider the provided metrics to be sound.
> 2. **How do you do the backward forecast?** For backward forecasting we reverse the input sequences and change the sign of winds and ocean currents. The second law of thermodynamics is obviously broken, however in the short-term timespan it still improves the performance of the gap filling as the diffusion lengthscale (including turbulent diffusion) is under the pixel size on these times.
> 3. **Augmentation strategy.** Yes, the same strategy is applied to train all models.
> 4. **Sentinel-1 Product details.** We use Sentinel-1 in Extra Wide (EW) mode to maximize coverage. Terrain radiometric correction on topography doesn’t affect sea surface, while the land surface is masked with zeros in terms to learn strictly sea ice dynamics.
> 5. **IIEE.** We agree that the definition and formula of this metric is important and added it in the Appendix, see line 867.
> 6. **Could you explain the filtration in other words again?** The filtration is realized via linear projection operator, which removes the part of additive noise with specific patterns (scalloping). The patterns were gathered from summer periods when there is no ice in the target region and were calibrated such that filtration would not modify the neutral image (the left one in Figure 1c).
> 7. **Why specific loss?** It is a heuristic solution. This combination enhanced the convergence in early experiments.
> 8. **How do you feed missing inputs to your models?** Missing values are filled with zeros for UNet and IAM4V. Other models impute the missing values with their intermediate forecast. We updated the manuscript to make it more clear (see Figure 3 and lines 341-343).
> 9. **The variation of missing data ratio.** The Sentinel satellite has power limitation, hence does not operate contagiously. We suggest that changes in the powering schedule might be the reason.
> 10. **Have you compared to a climatology?** We compared with climatologies, the proposed pipeline beats them by a wide margin. We provide our results bellow. However, we doubt that averaging over 9 years with lots of missing values provides a valuable baseline.
> ||MSE|IIEE@15|IIEE@30|IIEE@50|IIEE@75|
> |---|---|---|---|---|---|
> |monthly climatology|10.9|15.3|9.4|13.7|10.3|
> |weekly climatology|9.2|14.0|10.2|11.6|8.6|
> |our pipeline|6.8|10.0|9.0|9.2|5.3|
> 11. **Evaluated pipeline.** Forecast of the tested models were replaced with one of the DMVFN at dates with high uncertainty. Each model in the table is mixed with DMVFN except baselines and the DMVFN itself. Metrics without such mixture are provided in appendix at Table 8.

---

> > ### Comment · Reviewer_GA94 · 2024-11-28
> >
> > Dear authors,
> >
> > thanks a lot for responding to my concerns. Unfortunately, my skepticism towards the robustness of the presented results could not be addressed well enough to convince me otherwise. In addition, it appears the authors deem a lot of things "out of scope", but I would rather think that this work could really benefit a lot from another round of improvements making these things into scope. I will not raise my score, but I would nonetheless very much like to encourage you to continue to refine this work and, when further improved, to submit to a Journal.
> >
> > All the best, Reviewer GA94

---

### Official Review · Reviewer_GVYd · 2024-11-03

**Soundness:** 2
**Presentation:** 2
**Contribution:** 2
**Rating:** 6
**Confidence:** 4

**Summary:**

This work presents a sea ice forecasting approach that uses video prediction approaches applied to synthetic aperture radar (SAR) satellite imagery captured by Sentinel 1. The work examined the performance of a number of architectures for the video prediction task and uses an ensemble of four architectures to achieve uncertainty quantification.

**Strengths:**

The paper provides an approach to sea ice forecasting which is an important problem and explores the performance of several video prediction algorithms on this task. The authors also consider the problem of image artifacts and propose a projection based approach to eliminate image artifacts.

**Weaknesses:**

1. It is not clear what sea-ice parameters are considered in this work and how these parameters would be obtained from the SAR video streams. The authors should clearly state the parameters considered and describe how they are derived from SAR imagery.

2. The description of the architectures in Table 2 is not clear. What are the inputs and outputs in each configuration? Also, since the best performing system appears to be the rUNET system which is a SISO configuration, are the multiple inputs necessary or sources of potential error?

3. The IIEE metric is not explained in the paper. I believe it is the “integrated ice-edge error” which may be unfamiliar to other readers and should be introduced.

4. The data preprocessing step which involves learning a projection should be evaluated to validate the removal of artifacts? What is the computational complexity of this approach?



Minor Comments:
1. Typo - Line 145 “uncetrainty quantification”
2. A map of the area as would be useful in the main paper.

**Questions:**

1. Is the SAR video prediction and end in itself in this work?
2. Can  the performance of the  data preprocessing step be demonstrated quantitatively and also qualitatively by some example images?
3. Is there a way to incorporate ice-dynamics in this video prediction approach?
4. How sensitive would any approach be to location?

---

> ### Author Response · Authors · 2024-11-25
>
> We thank the reviewer for constructive feedback. We appreciate the acknowledgment of the importance and novelty of our work. Below, we address the specific concerns and questions raised in the review:
>
> ### Response to weaknesses:
> 1. **SAR parameters.** We directly use SAR HV and HH polarizations from Sentinel-1 extra-wide product. The preprocessing includes: conservative interpolation to the target resolution, normalization, and filtration. Processed SAR imagery is used as both input and target for neural networks.
> 2. **Models.** We have included more thorough models descriptions and the schematic visuals of the models and whole pipeline (please see Figure 3, lines 248-312 of the updated manuscript). In general, the input consists of described data channels sampled at sequential seven days (ten for IAM4VP), the outputs are predicted SAR images at the following three days. rUNet and all SISO models apply recurrently, the prediction is produced in the autoregressive manner (see Figure 3c).
> 3. **IIEE.** We added the definition of the integrated ice-edge error to the appendix (line 866)
> 4. **Preprocessing.** The projection operator is learnt once. Only the projecting should be evaluated to remove the artifacts. The computational complexity of evaluation a single image is O(h*w) where ‘h’ is height and ‘w’ is width in pixels. One can derive the complexity from the fact that the projection is a linear operator over the image space.
>
> ### Response to minor comments:
> 1. **Typos.** We fixed the typo, thanks for pointing them out!
> 2. **Map.** We added a map of the target area to the main body of the paper (see Figure 1a of the updated manuscript).
>
> ### Response to questions:
> 1. **Is the SAR video prediction and end in itself in this work?** SAR video prediction is an output of the developed pipeline. It was selected in collaboration with marine captains, who regard this data as a convenient and reliable source for assessing sea ice conditions. Additionally, the missing values imputation and uncertainty quantification are the byproducts of our forecasts and can be used as well.
> 2. **Preprocessing demonstration.** We added the examples of the filtration algorithm to the main body of the paper (see Figure 1c). The quantitative evaluation is out of scope of the paper, however we verify its importance via ablation study (appendix, Table 6)
> 3. **Is there a way to incorporate ice-dynamics in this video prediction approach?** The ice dynamics can be interpolated as specific parameterization of the neural ODE model or via addition of PINN-loss. However, the quality data on sea ice parameters, which is necessary to incorporate their dynamics, is unavailable in the target region due to its complex environmental conditions.
> 4. **How sensitive would any approach be to location?** We expect the pipeline to generalize well to other arctic rivers. However we acknowledge that the application over regions of the free sea surface might result in suboptimal performance in comparison with numerical models.

---

> > ### Comment · Reviewer_GVYd · 2024-11-27
> >
> > The authors have responded to my questions and made changes to the paper. I appreciate the effort of the authors to improve their paper and have adjusted my score upwards.

---

### Official Review · Reviewer_CmYN · 2024-11-04

**Soundness:** 2
**Presentation:** 2
**Contribution:** 2
**Rating:** 3
**Confidence:** 4

**Summary:**

The paper proposes different methods to forecast the sea ice extent in the gulf of Ob based a mix of Sentinel-1 data (radar), re-analysis data and interpolated weather stations.
The paper compares a slue of different methods, and aims at quantifying the uncertainty of the forecast with these methods. The different methods each produce a forecast, which is then used as an ensemble to quantify the uncertainty.

**Strengths:**

The beginning of the paper is well written, with a good problem statement and motivation. The importance of the work is well explained, and appears timely.

**Weaknesses:**

The paper compares 8 different methods to forecast the sea ice, but fails to introduce them. The author spend more time on the data preprocessing and filtration of S1 data, than on explaining what the actual models do. The only mention of the models are on line 79 to 94, but are very brief.

Overall, the paper lacks a significant analysis of the results. The results are shown briefly in table 3 and figure 3, but lack a deeper analysis. In the main text, there is no example of time series, nor map to show the uncertainty per pixel, nor interpolation output, or visuals to show the results, and help the reader in understanding the process.

The paper would profit massively from a schematic representation of the tasks.

I feel like the paper has potential, but the different sections have been given inappropriate weight. The paper would need a major restructuration, and overall would probably fit better in a longer format, such as a journal, where the details can be explained better, and the analysis performed at a deeper level. There are just too many moving parts to fit in this short format.

## Target
It is unclear to me how the target is produced. The authors mention "a target presentation of the forecasts" (line 213), but don't explain how they use Sentinel-1 to produce the target.

## Minor comments

Table 1: if the scale is supposed to be the scale of the product, then S1 has a scale of 10 meters, not 1km. The rescaled input is 1km, but so is GLORYS and the meteo stations
I would add the temporal resolution to this table to add a bit more information.

Line 206: "Sentinel-1 SAR images and GLORYS fields are interpolated bilinearly to match the input resolution (1 km)" using bilinear interpolation to resample Sentinel-1 from 10 meters to 1km is quite unconventional, usually downsampling is done with nearest neighbor or average.

Line 235: "up to 50 meters": as far as I know S1 resolution is 10 meters

hLine 257-262: this comment seems out of place.

Figure 3 is hard to read, is missing units, and pixelated.

Line 304: "nor noise" how do you make sure an image has no noise?

## Grammar comments

Line 079: "Our research employs advanced video prediction models, which include:" please rephrase, doesn't work with the bullet points

Line 240: "which lacks quality of ice data in the Gulf being mostly uncorrelated with other sources": unclear, rephrase

**Questions:**

c.f. weaknesses

---

> ### Author Response · Authors · 2024-11-25
>
> We thank the reviewer for valuable feedback and comments. We appreciate the recognition of the importance and potential of the work. Below, we address the specific concerns raised in the review:
>
> 1. **Model descriptions.** We added a section with enhanced model descriptions (lines 248 – 312) and a schematic figure to represent how autoregressive and non-autoregressive approaches manage data and missing values.
> 2.  **Representations of the task.** We include a schematic of the overall pipeline (see Figure 3a), the visualization of the target region, examples of the input SAR imagery to represent the problems of missing values and scalloping, and results of filtration to the main body of the paper.
> 3. **Paper structure.** We restructured the paper to shift focus on the task, applied models and the developed pipeline, and moved part of the data description into appendix.
> 4. **Target.** As a target we use HV polarization of SAR imagery. The same preprocessing applies both to input and target images: interpolation, normalization, filtration.
> 5. **Minor comments.** We fixed all typos. Both GLORYS and meteostations are daily-averaged to decrease computational burden of the models. SAR imagery is interpolated conservatively, line 206 was a typo. “Nor noise” meant that further apllication of the filtration won’t modify images; we removed these labels to avoid misunderstanding.

---

### Author Response · Authors · 2024-11-29
**General reply**

Dear Reviewers,

Thank you for your time and effort in reviewing our submission. We greatly appreciate your comments and suggestions, which have helped us improve the clarity and quality of the paper.

We are also grateful for the acknowledgment of the following aspects of our work:
 - Relevance of the task: The importance of satellite-based sea ice forecasting in Arctic navigation (reviewers CmYN, GVYd, GA94, CM4n).
 - Augmentation and preprocessing strategy: The physical and geometric augmentations and preprocessing tailored to this problem (reviewers GVYd, CM4n, GA94).
 - Uncertainty quantification: The introduction of an ensemble-based approach and model-switching for improving robustness (reviewer CM4n).
 - Thorough experiments: Comprehensive evaluation across multiple baselines and models (reviewer GA94).

Below, we summarize the main changes made to the manuscript (please, see the updated pdf) and address the common concerns raised across the reviews.

**Key Revisions and Updates**
- Expanded Model Descriptions and Pipeline Overview: Added detailed descriptions of the models used, including input/output configurations and how missing data is handled. These updates are supported by a schematic representation (Figure 3, lines 248-312).
- Improved Visualization: Included a map of the Gulf of Ob, examples of SAR imagery preprocessing, and filtering results (Figure 1a-c). These additions enhance understanding of the region and data challenges.
- Uncertainty Quantification: Clarified the role of the ensemble spread in providing uncertainty estimates and explained its use in the model-switching scheme (lines 386–400).
- Clarifications and Fixes: Addressed ambiguities about the IIEE metric (Appendix, line 866), preprocessing steps, and the handling of missing data.

**Addressing Common Points**
 - Novelty and Contributions: While our work builds on existing video prediction models, the introduction of the uncertainty-aware pipeline, ensemble methods, and task-specific augmentations represents a meaningful step forward in applying ML to geophysical problems. We’ve made these contributions more explicit in the revised text.
 - Practical Significance: We acknowledge that further evaluation in real-world scenarios would strengthen this point and see it as a natural next step.
 - Generalizability: Our approach is designed to work well in similar Arctic environments and could potentially be extended to other geophysical forecasting tasks. However, we recognize that additional adaptations might be needed for regions with different conditions, such as open ocean areas, where classical NWP ocean models may still be better.

**Acknowledgment of Updated Scores**

We thank Reviewers GVYd, CM4n and GA94 for revisiting the revised submission and Reviewers GVYd, CM4n for adjusting their scores. Your feedback and recognition of the changes made have been encouraging.
We kindly ask the remaining reviewers to consider the updates as well. We believe the revisions address the concerns raised and hope they meet your expectations.
Thank you once again for your thoughtful feedback and the opportunity to improve our work. We are happy to clarify further points if needed.

Best regards,

The Authors

---

### Meta-Review · Area_Chair_G7FF · 2024-12-23

**Metareview:**

The paper introduces a method called uncertainty-aware forecasting, which aims to predict sea-ice conditions in the Gulf of Ob as SAR images. This is achieved by applying existing deep learning video prediction models to multi-temporal, multi-band image data constructed from Sentinel-1 SAR images, weather observations, and GLORYS. The proposed method incorporates an ensemble-based approach for uncertainty quantification and utilizes a confidence-based model selection scheme to improve forecasting performance. The authors address challenges related to data irregularity and missing values. While the application is both interesting and important, the proposed uncertainty-aware model switching does not significantly outperform simpler approaches, such as UNet or rUNet. Consequently, the switching mechanism may be considered an incremental improvement over existing video prediction models.

**Additional Comments On Reviewer Discussion:**

The authors addressed some of the reviewers' concerns, leading to improved scores from certain reviewers. However, the rebuttal phase did not change the positions of reviewers CmYN, GA94, and CM4n regarding the lack of novelty in the proposed method.

---

### Decision · Program_Chairs · 2025-01-22

Reject